# A prospective longitudinal study with treated hypertensive patients in Northern Bangladesh (PREDIcT-HTN) to understand uncontrolled hypertension and adverse clinical events: A protocol for 5-years follow-up

**Ahmed Hossain** [1,2]*, **Gias Uddin Ahsan**[1], **Mohammad Zakir Hossain**[3,4], **Mohammad Anwar Hossain**[3], **Zeeba Zahra Sultana** [2], **Adittya Arefin** [2], **Shah Mohammad Sarwer Jahan**[3], **Probal Sutradhar**[3]

**1** Department of Public Health, North South University, Dhaka, Bangladesh, **2** Global Health Institute, North South University, Dhaka, Bangladesh, **3** Hypertension & Research Centre, Rangpur, Bangladesh, **4** TMSS Medical College, Bogra, Bangladesh

* ahmed.hossain@northsouth.edu

## Abstract

### Introduction

Uncontrolled hypertension is the most common cause of major adverse clinical events (MACE), such as myocardial infarction, strokes, and death due to CVDs, in both developed and developing countries. Western-led studies found that treated hypertensive adults with uncontrolled hypertension were more at-risk of all-cause and CVD-specific mortality than normotensives. The PRospEctive longituDInal sTudy of Treated HyperTensive patients of Northern-Bangladesh (PREDIcT-HTN) study principally aims to estimate the incidence of MACE in treated hypertensive patients and identify the determinants of MACE. The secondary objective is to find the prevalence of uncontrolled hypertension in treated hypertensive patients and the associated risk factors.

### Methods and analysis

The treated hypertensive patients were obtained from the Hypertension and Research Center (H&RC), Rangpur, Bangladesh, from January to December 2020. Based on the eligibility criteria, 2643 patients were included to constitute the PREDIcT-HTN cohort. Baseline data was retrieved from the H&RC registry, and five follow-up waves are planned yearly (2021–2025). A questionnaire will be administered at each follow-up visit on hypertension control status, behavioral factors, quality of life, dietary adherence, and high blood pressure compliance-related variables. The participant will be right censored if the patient develops MACE, death due to any cause, loss to follow-up, or at the end of the study. A proportional hazard model will identify the risk factors of MACE. Multinomial logistic regression analyses will be performed to determine the predictors of the hypertension control status by medication and dietary adherence after adjusting confounders.

**Data Availability Statement:** No datasets were generated or analysed during the current study. All relevant data from this study will be made available upon study completion.

**Funding:** This work was supported by North South University internal research grant (grant number CTRG-19-SHLS-37) after a peer review process and a similar amount was matched from Rangpur H&RC.

**Competing interests:** The authors have declared that no competing interests exist.

## Ethics and dissemination

The ethical approval for this study was obtained from the Institutional Review Board, North South University [Ref: 2019/OR-NSU/IRB-No.0902]. The participants will provide written consent to participate. The findings will be disseminated through manuscripts in clinical/academic journals and presentations at professional conferences and stakeholder communication.

## Introduction

Cardiovascular disease (CVD) is the principal cause of death in the world, with 17.3 million deaths per year and a projected increase to >23.6 million by 2030 [1]. Hypertension is the primary risk factor for CVDs globally, with uncontrolled hypertension being the most frequent cause of major adverse clinical events (MACE), including myocardial infarction, strokes, and death due to cardiovascular diseases in developed and developing countries [2]. It is critical to prevent, treat, and control hypertension to lower the risk of CVD events and the associated healthcare burden.

A recent systematic analysis of population-based studies from 90 countries showed that the global prevalence of adult hypertension was observed at 31.1%, and the proportions of awareness, treatment, control among treated patients, and control among all hypertensive patients were 46.5%, 36.9%, 37.1%, and 13.8%, respectively [3]. These results suggest the high burden of hypertension worldwide, with a high prevalence but low control rate. For chronic uncontrolled hypertension, for every 20-mmHg increase in systolic BP to > 115 mmHg or ten mmHg increase in diastolic BP to > 75 mmHg, the risk of vascular mortality doubles [4]. In a study of US adults, researchers discovered that treated hypertensive adults with uncontrolled hypertension had a higher risk of all-cause and CVD-specific death than normotensives [5]. Similarly, according to studies, Bangladesh has a prevalence of uncontrolled hypertension ranging from 25 to 50% [6, 7].

One of the key Sustainable Development Goals adopted by the World Health Assembly in 2013 was to lower the prevalence of raised blood pressure by 25% by 2025 [8]. Improvement in the management and control of hypertension will require understanding the factors that affect blood pressure control [9]. Because hypertension is usually asymptomatic, patients often do not seek care until significant damages have occurred, making effective control challenging [10]. A study conducted in China found that undetected, untreated, or uncontrolled hypertension patients were likely to develop stroke [11]. The risk was more among the individuals living in rural areas, with low education and occupation, and those with chronic conditions [11]. Additionally, a study conducted among US adults showed an increased proportion of patients with established atherosclerosis or at high risk of experiencing MACE by nearly 5-fold from 1-year to 4 of follow-up and a significant increase in risk among the treated hypertensive patients [12].

Multiple factors contribute to uncontrolled hypertension, despite variability among findings. Uncontrolled hypertension is caused by non-adherence to antihypertensive medication and the dietary approach to stop hypertension (DASH diet), high salt intake, smoking, physical inactivity, and overweight or obesity [13–15]. Those with uncontrolled hypertension had a higher risk of all-cause and CVD-specific death than those with normal blood pressure [15, 16]. Compliance with medication alone is only responsible for half of the antihypertensive drug failures, leading to suboptimal BP control at the population level, a high prevalence of

resistant hypertension, and an increased financial burden for healthcare [17, 18]. Moreover, age, sex, hypertension duration, and co-morbidities are also associated with uncontrolled hypertension [14–19].

The government of Bangladesh has demonstrated its commitment to NCD management and prevention by instituting a multi-sectoral action plan that will run from 2018 to 2025 [20]. The strategy intends to promote population health and strengthen the national hypertension management strategy by introducing evidence-based policies and activities. However, there is a dearth of longitudinal information focusing on the association of baseline characteristics and hypertension control status with the incidence of vascular events in the treated hypertensive population. Understanding it may greatly aid efforts to manage better and treat patients to prevent undesired events. Although few attempts, there is a dearth of evidence regarding cardiovascular disease occurrence among treated hypertensive patients at both global and national levels, including all adult individuals aged 18 years and above.

## Aim of the study

The three objectives of the PREDIcT-HTN study are (1) to determine the prevalence of uncontrolled hypertension among the treated hypertensive patients and the incidence of a major adverse cardiovascular event (MACE) at each follow-up and at the end of follow-up, (2) to find the risk factors of MACE after adjusting the confounders, and (3) to understand the change of quality of life of the treated hypertensive patients.

## Methods and materials

### Study design and setting

This is a prospective observational longitudinal study conducted at the Hypertension and Research Centre (H&RC), Rangpur, which is the largest hypertension management center in northern Bangladesh and one of the leading national institutes for hypertension research. A unique patient identifier is given to a new patient visiting the centre. It has over 25,000 registered patients and treats 50 patients daily on average. The overall approach of the PREDIcT-HTN has been directed to integrate research activities into H&RC's existing operational framework. We built and trained the PREDIcT-HTN administration, coordination, and implementation team (NACIT) at the center to support the study functions. For example, NACIT will assist with scheduling follow-up visits, conducting face-to-face interviews, taking physical measurements, retrieving registry data, and a three-step procedure to remind participants about follow-up. At the center, trained members of the H&RC staff will take on the role of operational experts to supervise the study's progress. A retrospective baseline data retrieval is planned from the H&RC registry during the first follow-up visit. Five follow-up waves are scheduled 12-monthly for five years from 2021 to 2025, each stop to be conducted from January–to March every year.

**Cohort identification and recruitment.** The target population is the 5874 adult hypertensive patients aged more than or equal to 18 years, residing in the northern part of Bangladesh and visiting the H&RC from January to December 2020. It was about 40% less than the usual number of patients visiting the center due to the COVID-19 pandemic in 2020 compared to 2019.

The study excluded 1352 unique patient identities because they were not diagnosed with hypertension from the list of registries by H&RC in 2020. Furthermore, patients having antihypertensive medication for less than six months, a history of cardiovascular disease, previously diagnosed with cancer with an effect on survival, and those with cognitive and mental

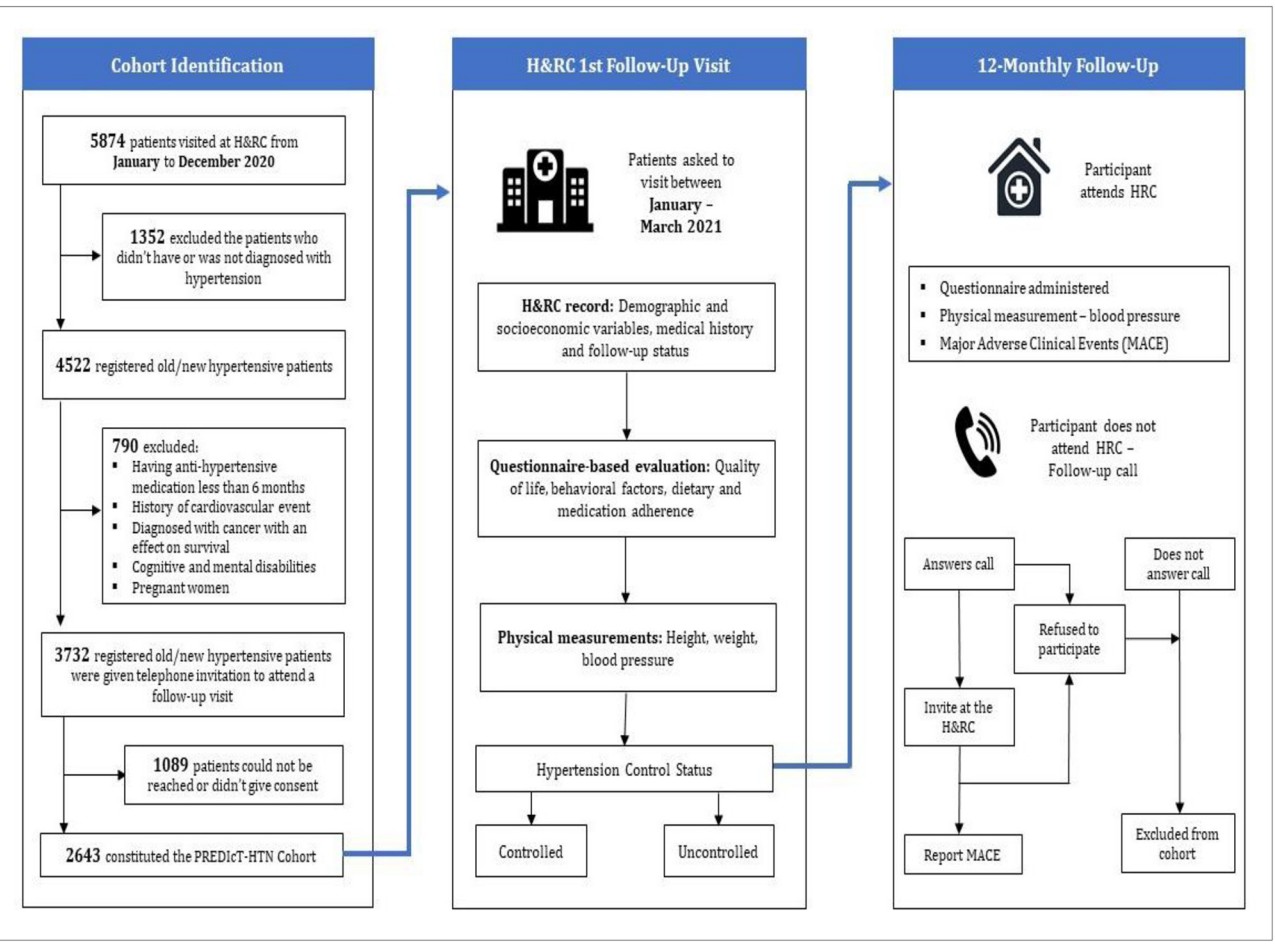

**Fig 1. The overall design for the PREDIcT-HTN study.**

disabilities were excluded. Finally, 3732 patients remained for cohort inclusion, and the flow-chart is given in Fig 1.

The 3732 patients were given invitations through telephone and mobile message to take part in which the study was discussed and asked to join at a scheduled appointment from January–to March 2021. The remaining 1086 patients could not be reached or refused to participate in the study. Finally, 2643 patients constituted the PREDIcT-HTN cohort. At the first follow-up visit, they were asked by the center reception staff whether they had received an invitation for the study and if they wanted to join. The respondents who accepted the invitation and visited H&RC were further continued in the follow-up process, and the baseline data was integrated. Patients who did not receive any invitation were not considered for the study. Finally, 2276 patients remained after the initial visit, accounting for a 14% non-response rate.

## Rationale for planned sample size

It is planned to follow up on a cohort of people recruited into the H&RC over some time as part of a study of poor therapy compliance on control of hypertension in patients from the H&RC. The subjects will be followed for a period of five years. According to the findings of a recent small-scale study, the annual incidence rate of adverse events (MCE) among

hypertension patients could be as high as 15%. We want to determine how many patients we will need at a 5% significance level and with an 80% power. The alternative hypothesis is that the annual incidence rate of adverse events (MCE) in the excellent therapy compliance group is less than 10%. Using the *epi.cohort()* function in *epiR* package provides the required sample size is 768. Based on the given reasoning, the planned sample size will have enough statistical power to meet any of the objectives of this study.

## Baseline data retrieval and first follow up visit

The unique IDs of the 2643 patients have been retrieved for the baseline assessment from the H&RC record registry. Data from 2276 patients were integrated from three sources during the initial visit: Retrieved baseline data, questionnaire-based administration, and physical measurements. The questionnaire has been designed to take approximately 20 minutes to complete and includes contact details of the respondents, behavioral factors, quality of life, dietary adherence, and high blood pressure therapy compliance. Physical measurements have height, weight, and blood pressure records.

## Yearly follow-up plan

Participants will be asked to return to the H&RC every 12 months for the next four years. The initial questionnaire will be administered at each appointment, along with updated contact information, and physical measurements will be taken at each follow-up visit. Telephonic, mobile messages and postal invitations will be sent before each follow-up visit. A follow-up call will be made if a respondent does not appear for their planned appointment. If he answers the phone, they will again be invited to come to the H&RC with a new appointment date. If they do not come on the rescheduled day, they will be asked to report any adverse cardiovascular events over the telephone. The administration and reminder procedures for every follow-up will be identical.

End of follow-up will occur in four instances: at 5-year, if a patient reaches a composite cardiovascular disease as myocardial infarction or ischemic heart disease including angina or stroke or transient ischemic attack, death due to cardiac or non-cardiac causes or loss to follow-up. An example with seven hypertensive patients is given in Fig 2. If any other event occurs during follow-up, only the first event will be included, and the participant will be censored. The information on the incidence of cardiovascular disease will be taken from the primary diagnosis of the hospital discharge card following the event. The cause of death will be ascertained based on the death certificate or through telephonic conversation with the patients' attendance. Even though the patients will be followed up annually as part of the trial, they will come to the center as their physician directs. In addition, the measures used to assess the quality of life, antihypertensive medication compliance, and nutritional adherence all involve recalling the previous seven days on the day of data collection. As a result, there is no issue with the follow-up interval. The duration of follow-up has less of an impact on loss to follow-up for comparable analysis.

Additionally, the patients will be contacted by telephone every six months to verify whether they have been diagnosed with any target event of this study. Also, all participants will be asked to immediately notify the research team should they be diagnosed with any cardiovascular event at any time during the life of the study.

## Outcome measurements

**Primary outcome.** The incidence and predictors (hypertension status, demographic, behavior, quality of life, dietary adherence, and high blood pressure therapy compliance) of

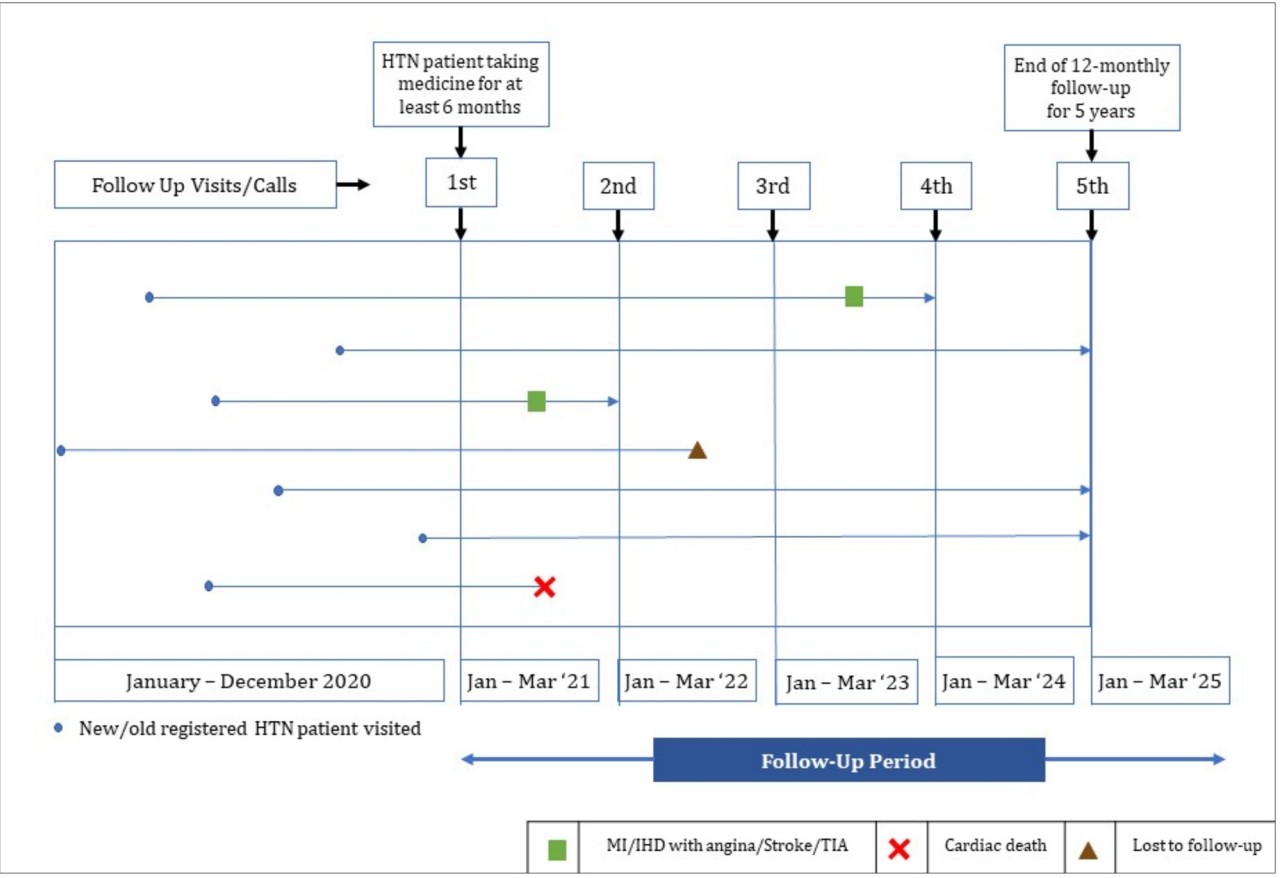

**Fig 2. Inception and follow-up example of 6 respondents of the PREDIcT-HTN study.**

major adverse clinical events (MACE) in the PREDIcT-HTN cohort at a 5-years follow-up period. MACE is a composite of cardiovascular event occurrence or death due to cardiovascular disease [21].

**Key secondary outcome.** The prevalence of uncontrolled hypertension among treated hypertensive patients at each follow-up visit. Following the 2020 International Society of Hypertension Global Hypertension Practice Guideline and will be using the following definitions for the study [22]:

*Hypertension*: Systolic blood pressure ≥140 mmHg and/or diastolic blood pressure ≥90 mmHg

*Grade I Hypertension*: Systolic blood pressure between 140–159 mmHg and/or diastolic blood pressure between 90–99 mmHg

*Grade II Hypertension*: Systolic blood pressure ≥160 mmHg and/or diastolic blood pressure ≥100 mmHg

Secondary outcomes:

1. Association between dietary adherence and high blood pressure therapy compliance with hypertension control status of the participants (controlled and uncontrolled grade I and II).

2. The incidence and predictors of all-cause mortality of the respondents of the PRE-DIcT-HTN cohort.

3. Quality of life of the treated hypertensive patients.

4. The rate of attrition (not having a final visit at 12 months) and rates of several categories of attrition (mortality, withdrawal from the study, and loss to follow-up without identifiable cause).

## Evaluation and instrumentation

### H&RC registry record

The investigators have confidentially been retrieved data from the H&RC registry on participants' demographic and socio-economic determinants (age, sex, residence, marital status, education level, occupation, household income, and commute to work), medical history (diagnosed comorbidities, age of first hypertension diagnosis, years of hypertension diagnosis and a number of medication), follow-up status of the hypertensive patients at the center and cause of irregularity. Using secured systems, these data will be transferred to the coordinating team to facilitate the updated contact of participants, merge with the questionnaire-based data and physical measurements, tracking of participants' follow-up appointments throughout the study. Regular follow-up is defined as participants who attended 80% of the scheduled follow-up as per the record book and were considered irregular if less than 80% of the scheduled follow-up.

**Blood pressure measurement.** Blood pressure will be measured in the left arm three times with a 3 minutes interval between each measurement. The patient will remain in a sitting position for at least 10 minutes before the first evaluation. If one of the three measurements is quite different, another measure will be taken. The recorded blood pressure will be the average of the three closer measurements. A cuff in accord with the diameter of the arm and a mercury sphygmomanometer, previously calibrated, will be used.

### Anthropometry

Evaluation of anthropometric measurements (weight and height) will be performed with the patient fasting, shoeless, and wearing a hospital gown by the procedures described by 'The International Society for the Advancement of Kinanthropometry (ISAK)' [23]. For weight measurement, the patient will be asked to stand on the center of the scale without support, with the arms hanging freely to the sides and with the weight distributed evenly on both feet. A mechanical column scale (SECA 700) with a capacity of 220 kg and precision of 0.05 kg will be used, and the weight will be recorded to the nearest 100 g. Height will be measured with a stadiometer SECA 220, with the subject standing with the feet together and the heels, buttocks, and upper part of the back-projected on the same vertical plane. Measurement will be taken at the end of a deep inward breath, placing the headboard firmly down on the vertex and recorded to the nearest millimeter.

### Behavioral factors

Data will be collected on smoking status (ever or never smokers). Participants who report to smoke at least 100 cigarettes in their lifetime and who smoked either every day or some days will be classified as ever smokers (current and ex-smokers) [24]. Additionally, the self-reported information on extra salt consumption in the everyday meal will also be documented.

## Quality of life

The validated *WHOQOL-BREF* will measure the quality of life (QOL) [25, 26]. It consists of 24 items to assess the perception of quality of life in four domains, including physical health, psychological, social relationships, and environment, and two articles on overall QOL and general health. The categories of the scale are (1) very poor, (2) poor, (3) neither poor nor good, (4) good, and (5) very good. Following the scoring guidelines, the domain scores will be transformed into a linear scale between 0 and 100. A higher score will indicate a better QOL. According to the standardized instructions, the questions will be read out to respondents as the assessment be interviewer-administered.

## High blood pressure therapy scale

The 14-item *Hill-Bone Compliance to High Blood Pressure Therapy Scale* (HB-HBP) in Bengali will be used. HB-HBP is a **14-item** scale that assesses patient behaviors for three important behavioral domains of high blood pressure treatment (i.e., the three (3) sub-scales): appointment keeping (3-items), diet (2-items), and medication adherence (9-items) [27]. Each item is scored on a 4-point Likert scale, with a score of 4, meaning the highest level of compliance. The maximum and minimum scores are 56 and 14, respectively, for all 14 items. The top scores for medication adherence, sodium intake, and appointment keeping subscales are 36, 12, and 8. The higher the scores, reflects poorer adherence to anti-hypertensive therapy. It can both be self-administered or interviewer-administered. The mode of data collection for this study would be interview administered. It is helpful to assist at every visit because it is beneficial in planning and to implement effective individualized HBP care.

## Dietary adherence

Participants' dietary assessment will be done through a modified form of *food frequency questionnaire* (FFQ) from which a *DASH score* was constructed based on eight food groups or nutrients for which consumption should be increased (fruits, vegetables, nuts and legumes, low-fat dairy, whole grains) or reduced (sodium, sweetened beverages, red and processed meats) [28]. Consumption of each dietary component will be divided into three groups, and participants' intakes were assigned 0–2 points. Component scores will be summed, and an overall DASH score ranging from 8–to 40 was calculated. Afterward, DASH scores will be collapsed to tertiles, and dietary quality was classified into low, medium, and high tertiles based on adherence to dietary recommendations. In our study, low and medium dietary adherence will be considered unhealthy.

## Ethical and safety issues

The ethical approval for this study was obtained from the Institutional Review Board, North South University [Ref: 2019/OR-NSU/IRB-No.0902]. Informed written consent of each participant will be taken. All ethical principles of the Helsinki declaration were maintained throughout the study.

The research implementation team conducting face-to-face interviews will provide all the required measures before, during, and after data collection, following a standard protocol [29]. The enumerators will be supplied with all COVID-19 safety precautions (e.g., wearing a mask, hand sanitizers, social distancing, etc.) with a reminder of the general guidance and protocol daily during the data collection period.

Participants can leave the study at any time and without giving a reason. Participants are urged to continue providing information on any critical clinical event that occurs without the

provision of a questionnaire or physical assessments to preserve maximal data capture. If a participant still decides to withdraw from the study, they must provide permission to the research team to keep and utilize the previously provided data.

The cohort study's findings will be disseminated through peer-reviewed articles, presentations at scientific events, and local stakeholder conferences. The researchers may also use ad hoc meetings, occurrences, or press releases to share aggregated data with the general public and clinical experts.

## Data screening and analysis plan

### Data capture and management

The data from the PREDIcT-HTN study cannot be stored anonymously since the study involves respondents repeatedly visiting the research center. A study participant code will be provided for each included patient. The unique patient ID of the H&RC and the study participant code cannot be derived from one another. The only means of linking the collected data is through a linking table, access to which is limited to the PREDIcT-HTN data managers and a selected few limited study personnel. Data will be captured using two dedicated computers to store information in a database structure. A soft copy of the data will be password protected and managed using the WINDOWS programs.

### Principles

The data-entry team will ensure that all data needed are collected. If the core data are missing, clarification requests will be immediately sent to the attending physician. Data cleaning, including logical, outlier, and variable engineering, will be performed. The original variables will be transformed if needed, such as converting continuous variables to categorical variables and combining multiple variables into single variables for information integration. We will analyze the data using R statistical software Version 3.6.3. A significant effect for the main products will be considered using a 95% confidence interval of risk ratios and hazard ratios.

### Follow-up data analyses

The cohort profile will be summarized using baseline data. Descriptive analyses of the first follow-up visit will be presented by hypertension control status (controlled and uncontrolled grade I and grade II) of the treated hypertensive patients. Discrete variables will be summarized by frequencies and percentages. Percentages will be calculated according to the number of patients for whom data is available. Continuous variables will be summarized by the mean +/- SD or median with 25% to 75% quartiles. Statistical inference of continuous variables will be performed using student's t-test or Wilcoxon rank-sum tests as appropriate. The Pearson's Chi-square test or Fisher exact test will be used, as appropriate, for categorical data.

Furthermore, multinomial logistic regression analysis and proportional hazard model will be performed to determine the predictors of uncontrolled hypertension among treated hypertensive patients. The exponentiated regression coefficient will be used, presented as risk ratio (RR) or hazard ratio (HR) with a 95% confidence interval. Sensitivity analyses of the predictors of the hypertension control status will be performed by age ($>= 45$ years and $<45$ years) and gender (male and female). The MACE will be reported as cumulative incidences (proportion of patients experiencing an event) and incidence rates (events/100 person-years) at each yearly follow-up (2nd to 5th). The attrition rates will be calculated according to the number of patients for whom data is available.

### End-of-study analyses

Individuals will be right-censored if the participant experiences a major cardiovascular event, dies due to any cause, or is lost to follow-up, considering the date of the last contact. The cumulative and individual components of MACE will be reported as cumulative incidences (proportion of patients experiencing an event) and incidence rates (events/100 person-years) at five years. The Cox proportional hazards model will estimate the hazard ratio (HR) of event occurrence. Unadjusted, age-sex adjusted, and multivariable-adjusted HRs of MACE in the PREDIcT-HTN cohort participant will be calculated to investigate the association between an explanatory variable and MACE incidence rates. Subgroup analyses will be stratified by age, gender, and the presence of co-morbidities.

### Questionnaire development with patients' involvement

In Northern Bangladesh, 15 hypertension patients from H&RC gave comments on the initial draft of the questionnaire after participating in pilot testing. They were satisfied with the questionnaire's content, emphasizing the importance of incorporating questions about their quality of life, support needs, and medical outcomes. Workshops and medical camps will convey summaries of study findings to study participants.

### Limitation of the study

We anticipate several limitations in the study. We expect some missing data from patients, and we expect some patients to seek health care outside of the H&RC during the follow-ups. However, our analyses can accommodate missing data resulting from the absence of patients. We will use multiple imputation methods developed for big data to include these patients in the comments. Unobserved changes may occur over time in the study during the current pandemic, making it difficult to follow up on the patients. We will also run sensitivity assessments to draw the conclusion if needed.

## Conclusion

The 5-year PREDIcT-HTN study will use a longitudinal design to estimate the incidence of major adverse clinical events (MACE) and investigate its associated variables among treated hypertension patients. Fewer studies in Bangladesh have followed up with hypertensive patients and evaluated the risk of cardiovascular events occurrence and mortality. The project enrolled 2643 persons after the first visit and collected data on demographics, behavior, quality of life, food habits, and high blood pressure therapy compliance. The findings will aid in identifying at-risk populations and testing future interventions aimed at improving health outcomes for both young and old persons, which could have translational implications. Nevertheless, the PREDIcT-HTN cohort is culturally appropriate and will inform physicians, academics, and policymakers about the gaps in Bangladeshi adults' hypertension care.

## Supporting information

**S1 Checklist. STROBE statement—Checklist of items that should be included in reports of cohort studies.**
(DOCX)

## Author Contributions

**Conceptualization:** Ahmed Hossain, Gias Uddin Ahsan, Mohammad Zakir Hossain, Mohammad Anwar Hossain, Adittya Arefin, Shah Mohammad Sarwer Jahan.

**Funding acquisition:** Ahmed Hossain, Gias Uddin Ahsan.

**Investigation:** Mohammad Anwar Hossain.

**Methodology:** Ahmed Hossain.

**Project administration:** Mohammad Zakir Hossain, Mohammad Anwar Hossain, Zeeba Zahra Sultana.

**Resources:** Mohammad Zakir Hossain, Shah Mohammad Sarwer Jahan, Probal Sutradhar.

**Supervision:** Ahmed Hossain.

**Visualization:** Zeeba Zahra Sultana.

**Writing – original draft:** Ahmed Hossain, Zeeba Zahra Sultana, Adittya Arefin.

**Writing – review & editing:** Ahmed Hossain, Gias Uddin Ahsan, Mohammad Zakir Hossain, Mohammad Anwar Hossain, Zeeba Zahra Sultana, Adittya Arefin, Shah Mohammad Sarwer Jahan, Probal Sutradhar.

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
