## [Decision Letter · Decision Letter 0]

29 Apr 2022

PONE-D-22-07048Prospective longitudinal study with treated hypertensive patients in Northern-Bangladesh (PREDIcT-HTN) to understand uncontrolled hypertension and adverse clinical events: A protocol for 5-years follow-upPLOS ONE

Dear Dr. Ahmed Hossain,

Thank you for submitting your manuscript to PLOS ONE. After careful consideration, we feel that it has merit but does not fully meet PLOS ONE’s publication criteria as it currently stands. Therefore, we invite you to submit a revised version of the manuscript that addresses the points raised during the review process. Please submit your revised manuscript by Jun 03 2022 11:59PM. If you will need more time than this to complete your revisions, please reply to this message or contact the journal office at plosone@plos.org. Please include the following items when submitting your revised manuscript:A rebuttal letter that responds to each point raised by the academic editor and reviewer(s). You should upload this letter as a separate file labeled 'Response to Reviewers'.A marked-up copy of your manuscript that highlights changes made to the original version. You should upload this as a separate file labeled 'Revised Manuscript with Track Changes'.An unmarked version of your revised paper without tracked changes. You should upload this as a separate file labeled 'Manuscript'.

We look forward to receiving your revised manuscript.

Kind regards,

Professor Hafiz T.A. Khan, Ph.D, CStat

Academic Editor

PLOS ONE

Journal Requirements:

- https://www.nature.com/articles/s41598-018-27377-2

- https://bmccardiovascdisord.biomedcentral.com/articles/10.1186/s12872-019-1091-6

- https://pubmed.ncbi.nlm.nih.gov/29447174/

The text that needs to be addressed involves the Introduction

In your revision ensure you cite all your sources (including your own works), and quote or rephrase any duplicated text outside the methods section. Further consideration is dependent on these concerns being addressed.

4. Thank you for stating the following in the Funding Section of your manuscript: 

"This work was supported by North South University internal research grant and a similar amount was matched from Rangpur H&RC."

We note that you have provided funding information. However, funding information should not appear in the Funding section or other areas of your manuscript. We will only publish funding information present in the Funding Statement section of the online submission form. 

"This work was supported by North South University internal research grant (grant number CTRG-19-SHLS-37) after a peer review process and a similar amount was matched from Rangpur H&RC."

Reviewers' comments:

Reviewer's Responses to Questions

**Comments to the Author**

1. Does the manuscript provide a valid rationale for the proposed study, with clearly identified and justified research questions?

Reviewer #1: Yes

Reviewer #2: Yes

2. Is the protocol technically sound and planned in a manner that will lead to a meaningful outcome and allow testing the stated hypotheses?

Reviewer #1: Yes

Reviewer #2: Yes

3. Is the methodology feasible and described in sufficient detail to allow the work to be replicable?

Reviewer #1: Yes

Reviewer #2: Yes

4. Have the authors described where all data underlying the findings will be made available when the study is complete?

Reviewer #1: Yes

Reviewer #2: Yes

5. Is the manuscript presented in an intelligible fashion and written in standard English?

Reviewer #1: Yes

Reviewer #2: Yes

6. Review Comments to the Author

You may also provide optional suggestions and comments to authors that they might find helpful in planning their study.

Reviewer #1: Abstract:

• Method:

o What does it mean by “treated hypertensive patients”? A little more clarification in the abstract would be great.

o “The participant will be right censored if the patient develops MACE or death due to any cause, lost to follow-up or at end of the study”. – Is this an extra “Or” in this sentence?

Introduction:

• Why is this study important? We already know most of the risk factors of hypertension or CVD now. The authors can add some additional justifications in the Introduction - what is so unique about the study? And why do they want to conduct search research in Bangladesh?

Methods:

•Study design and setting:

o Is there any justification for – why you want to follow up with the enrolled participants 12 monthly? The gap between the follow-up periods is long, but there will also be more recall bias, death, and loss to follow-up (refuse, absent, etc.). How will you be going to manage all of these?

o Please mention in the write up clearly – who will be excluded from the study? Figure 1 has all the details, but better to mention them in the write-up.

• Rationale for planned sample size:

o What percentage did the authors consider for the loss to the follow-up group?

Informed Consent form:

•Did the author plan to collect informed consent in every follow-up visit? If not, then the author should clearly mention the number of follow-up visits and how many years the participants will be followed-up?

Reviewer #2: Thank you for submitting your proposal to conduct this important research in northern Bangladesh. Certainly, there is a dearth of data regarding the hypertension control status and major adverse events in this part of the country as well as in Bangladesh. There are a few issues to be addressed to improve the quality and robustness of the study and make it more lucid.

There are a few instances where it is written that it will be done. However, what I believe is that these stages have already been done as per the time line set in the proposal. For example, in the abstract, it is written that the baseline data will be retrieved. Likewise, the participants have already given their consent and will not provide it. Please find these and correct them.

The first objective of the study stated must include the aim of investigating the 5-year incidence of MACE (Major Adverse Clinical Events).

3. An additional value could be added by taking the rate of attrition (not having a final visit at 12 months) and rates of several categories of attrition (mortality, withdrawal from the study, transfer to non-study clinics, and loss to follow-up without identifiable cause) into consideration as a secondary outcome. Please include it in the analysis plan as well.

As per the follow-up design of the study, the death of the participant shall be recorded based on the death certificate or through telephonic conversation with the patient’s attendance. It might not be robust, but it will surely be a transparent and executable study plan.

7. PLOS authors have the option to publish the peer review history of their article (what does this mean?). If published, this will include your full peer review and any attached files.

Reviewer #1: No

Reviewer #2: No

---

## [Author Response · Author response to Decision Letter 0]

9 May 2022

Reviewer #1: 

1. Abstract:

o Method:

a. What does it mean by “treated hypertensive patients”? A little more clarification in the abstract would be great.

b. “The participant will be right censored if the patient develops MACE or death due to any cause, lost to follow-up or at the end of the study”. – Is this an extra “Or” in this sentence?

Authors: Thank you very much for your comment. The term "treated hypertensive patients" refers to patients with hypertension receiving antihypertensive treatment. Patients are ignorant about their high blood pressure and thus do not take the necessary treatment in many instances. Other factors could include a poor socioeconomic situation, reliance on alternative medicine, and so on. However, the mentioned term is well-recognized in previous publications as well. A citation from a Q1 journal is given below:

R. Izzo et al., “Development of Left Ventricular Hypertrophy in Treated Hypertensive Outpatients,” Hypertension, vol. 69, no. 1, pp. 136–142, Jan. 2017, doi: 10.1161/HYPERTENSIONAHA.116.08158.

On the second point, as MACE is the primary outcome, it has been separated with an extra "or". It appears that there is a grammatical mistake, which is corrected.

2. Introduction:

o Why is this study important? We already know most of the risk factors of hypertension or CVD now. The authors can add some additional justifications in the Introduction - what is so unique about the study? And why do they want to conduct search research in Bangladesh?

Authors: Thank you very much for your observations. The study's objective is to estimate the incidence of MACE and identify the associated factors related to uncontrolled hypertension. It is relevant because this information has never been investigated in northern Bangladesh, which partly represents the rural population of Bangladesh. The relationship between socio-demographic factors, anti-hypertensive compliance therapy, medication and dietary adherence, and hypertension control status would help identify implementation and policy gaps regarding lifestyle modification awareness and approaches. We are aware of the risk factors for hypertension and CVD; however, we do not intend to investigate them. 

3. Methods:

o Study design and setting:

a. Is there any justification for – why you want to follow up with the enrolled participants 12 monthly? The gap between the follow-up periods is long, but there will also be more recall bias, death, and loss to follow-up (refuse, absent, etc.). How will you be going to manage all of these?

b. Please mention in the write up clearly – who will be excluded from the study? Figure 1 has all the details, but better to mention them in the write-up.

Authors: Thank you again for the comments. Even though the patients will be followed up annually as part of the trial, they will come to the center as their physician directs. In addition, the measures used to assess the quality of life, antihypertensive medication compliance, and nutritional adherence all involve recalling the previous seven days on the day of data collection. As a result, there is no issue with the follow-up interval. The duration of follow-up has less of an impact on loss to follow-up for comparative analysis. Additionally, the mechanism of re-inviting patients is described in detail. It also appears that the number of deaths will vary because of this gap.

The end of follow-up or the individuals who will be excluded is described in the "Yearly Follow-up Plan" section (Page 8, Line 208). Furthermore, the definition of MACE (major adverse cardiovascular event) is stated in the "Introduction" (Page 4, Line 74).

o Rationale for planned sample size:

a. What percentage did the authors consider for the loss to the follow-up group?

Authors: Thank you very much for your comment. The required sample size is 768, whereas we aimed to recruit 2643. The planned sample size will have enough statistical power to meet any of the objectives of this study, even if the loss to follow-up rate is between 15-20% in each of the five visits. During the first follow-up, 2276 of the 2643 respondents visited the center, accounting for an about 14% non-response rate within the acceptable margin. This information is now added in the manuscript.

4. Informed Consent form:

o Did the author plan to collect informed consent in every follow-up visit? If not, then the author should clearly mention the number of follow-up visits and how many years the participants will be followed-up?

Authors: Thank you very much for your suggestion. We already have mentioned these in our informed consent form and took the consent.

Reviewer #2: 

1. There are a few instances where it is written that it will be done. However, what I believe is that these stages have already been done as per the time line set in the proposal. For example, in the abstract, it is written that the baseline data will be retrieved. Likewise, the participants have already given their consent and will not provide it. Please find these and correct them.

Authors: Thank you very much for your observation. We have edited the manuscript accordingly.

2. The first objective of the study stated must include the aim of investigating the 5-year incidence of MACE (Major Adverse Clinical Events).

Authors: Thank you very much for this comment. We have included it in the manuscript.

3. An additional value could be added by taking the rate of attrition (not having a final visit at 12 months) and rates of several categories of attrition (mortality, withdrawal from the study, transfer to non-study clinics, and loss to follow-up without identifiable cause) into consideration as a secondary outcome. Please include it in the analysis plan as well.

Authors: Thank you very much for your suggestion. The attrition rate can be calculated as per our analysis plan; thus, we have included it in the paper.

4. As per the follow-up design of the study, the death of the participant shall be recorded based on the death certificate or through telephonic conversation with the patient’s attendance. It might not be robust, but it will surely be a transparent and executable study plan.

Authors: Thank you very much for such a detailed observation. We have corrected it accordingly.

---

## [Editor Report · Decision Letter 1]

18 May 2022

A prospective longitudinal study with treated hypertensive patients in Northern Bangladesh (PREDIcT-HTN) to understand uncontrolled hypertension and adverse clinical events: A protocol for 5-years follow-up

PONE-D-22-07048R1

Dear Authors,

We’re pleased to inform you that your manuscript has been judged scientifically suitable for publication and will be formally accepted for publication once it meets all outstanding technical requirements.

Kind regards,

Hafiz T.A. Khan, Ph.D, CStat

Academic Editor

PLOS ONE
---

## [Editor Report · Acceptance letter]

20 May 2022

PONE-D-22-07048R1 

A prospective longitudinal study with treated hypertensive patients in Northern Bangladesh (PREDIcT-HTN) to understand uncontrolled hypertension and adverse clinical events: A protocol for 5-years follow-up 

Dear Dr. Hossain:

I'm pleased to inform you that your manuscript has been deemed suitable for publication in PLOS ONE. Congratulations! Your manuscript is now with our production department. 

Kind regards, 

on behalf of

Professor Hafiz T.A. Khan 

Academic Editor

PLOS ONE